# Using Gd-Enhanced β-NaYF$_4$:Yb,Er Fluorescent Nanorods Coupled to Reduced TiO$_2$ for the NIR-Triggered Photocatalytic Inactivation of *Escherichia coli*

**Huang Zhou and Fengjiao He ***

State Key Laboratory of Chemo/Biosensing and Chemometrics, College of Chemistry and Chemical Engineering, Hunan University, Changsha 410082, China; huangzhou527@hnu.edu.cn

* Correspondence: fengjiaohe@hnu.edu.cn

**Abstract:** β-NaYF$_4$:Yb,Er,Gd fluorescent nanorods were successfully coupled to a reduced TiO$_2$ (UCNPs@R-TiO$_2$) nanocomposite and applied to visible-light catalytic sterilization under 980 nm near-infrared (NIR) light illumination. The UCNPs (β-NaYF$_4$:Yb,Er,Gd) absorb the NIR light and emit red and green light. The visible light can be absorbed by the R-TiO$_2$ (Eg = 2.8 eV) for the photocatalytic reaction. About 98.1% of *Escherichia coli* were effectively killed upon 12 min of NIR light irradiation at a minimum inhibitory concentration (MIC) of 40 μg/mL UCNPs@R-TiO$_2$ nanocomposite. The bactericidal properties were further evaluated by matrix-assisted laser desorption/ionization time-of-flight mass spectrometry (MALDI-TOF MS) analysis. We found that the high bactericidal activity was due to the synergistic effect between the UCNPs and R-TiO$_2$. Moreover, the UCNPs show excellent upconversion luminance properties, and the introduction of visible-light-absorbed R-TiO$_2$ nanoparticles (2.8 eV) was conducive to the efficient separation and utilization of photogenerated electron-hole pairs.

**Keywords:** β-NaYF$_4$:Yb,Er,Gd nanorods; reduced TiO$_2$ nanoparticles; near-infrared light; visible light; *Escherichia coli*

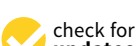



## 1. Introduction

Bacterial infections can cause great harm to public health and have attracted considerable research attention from scientists. To date, many traditional bactericidal agents manifest a wide range of potential applications in antibacterial disinfection [1,2]. However, ordinary bactericidal agents have been shown to induce risks to the environment and encourage antibiotic resistance [3]. Contemporarily, the photocatalytic sterilization created by utilizing some reactive species produced under ultraviolet or visible light illumination has become increasingly valuable [2,4–6]. In short, photocatalytic nanocomposites harvest optical energy to produce positive and negative charge carriers that are involved in photoredox reactions [7–9]. It has been reported that TiO$_2$ is one of the most representative and dominant photocatalysts on account of its non-toxicity and chemical stability. It, however, has a wide-band energy gap of 3.2 eV, which endows it with a high light absorption capacity within the ultraviolet (UV) light region to enhance photocatalytic efficiency with hardly any absorption within the visible region [10]; UV light and visible light account for only 5% and 45% of sunlight, respectively [10,11]. To improve the efficiency of sunlight, it would be interesting to develop TiO$_2$-based materials that can absorb visible light. The ion doping or morphology-engineering of TiO$_2$ nanoparticles can narrow their energy bandgap, resulting in the efficient absorption of visible light. For example, two-dimensional reduced TiO$_2$ nanosheets with an energy bandgap of 2.86 eV have been used to efficiently inactivate bacteria under visible light irradiation [11]. Hydrothermally synthesized TiO$_2$ nanosheets doped with N, C, and/or S also exhibit visible light absorption capacities [12–16]. However, ultraviolet and visible light can injure healthy tissue and display a short light penetration

depth in the human body, both of which hinder their further use in in vivo antibacterial applications [17].

Recently, near-infrared (NIR) light-induced upconversion particles have been introduced to photocatalyst sterilization systems. The upconversion particles can absorb low NIR photons, convert them to higher energy photons, and emit ultraviolet or visible light [18,19]. Coupling $TiO_2$ with upconversion particles can remarkably extend the utilization of the whole solar spectrum [20]. It has been suggested that β-$NaYF_4$ is the most ideal upconversion matrix for the doping of different lanthanide ions, on account of its high refractive index and transparency [21]. It may act as an intermedium for transferring NIR light energy to UV-Vis light that can be absorbed by the $TiO_2$ nanoparticles to produce oxidative holes ($h^+$) and reductive electrons ($e^-$). Effective $e^-$-$h^+$ pairs are able to react with $O_2$, OH, and $H_2O$ in a mixed solution to generate various reactive species that are helpful for sterilization [22,23]. For instance, β-$NaYF_4$:Yb,Er nanomaterial has been widely used in biological imaging analysis and photocatalytic applications. However, its luminous intensity is not satisfactory [24,25].

In this study, Gd-enhanced β-$NaYF_4$:Yb,Er,Gd fluorescent nanorods with a high fluorescence intensity were coupled to reduced $TiO_2$ nanoparticles with excellent visible light absorption abilities (UCNPs@R-$TiO_2$ nanocomposite) using electrostatic assembly (Figure 1). As expected, the created UCNPs@R-$TiO_2$ nanocomposite exhibits an effective photocatalytic sterilization performance against *Escherichia coli* (*E. coli*) under 980 nm NIR light illumination. The in vitro cellular cytotoxicity and antibacterial performance of the obtained UCNPs@R-$TiO_2$ were also evaluated by MTT (3-(4,5)-dimethylthiahiazo (-z-y1)-3,5-di-phenytetrazoliumromide) assay and MALDI-TOF MS analysis, respectively.

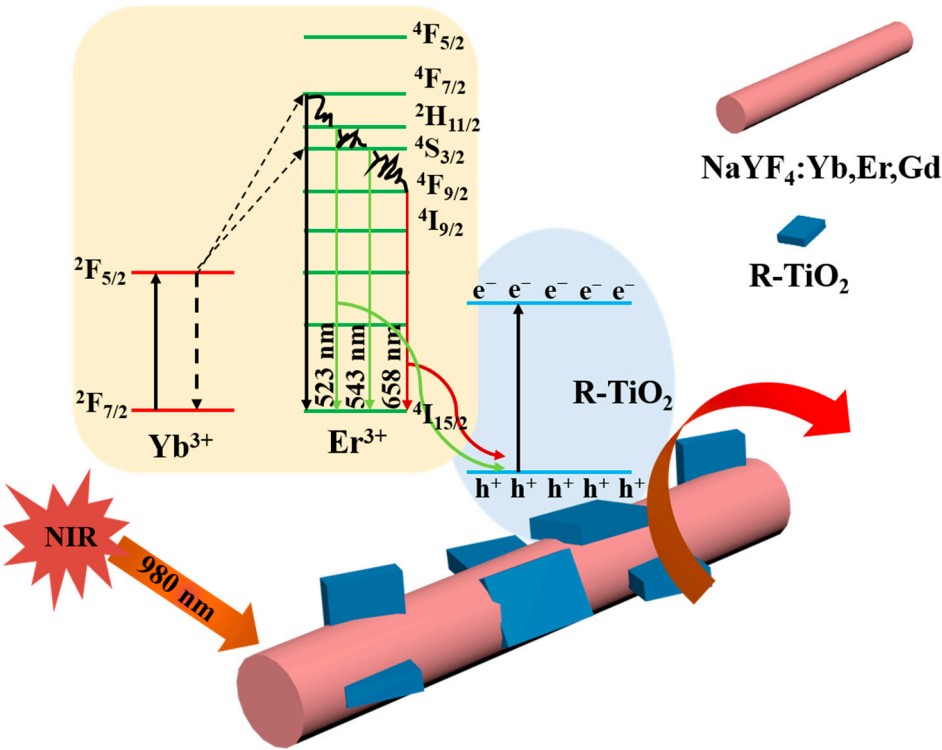

**Figure 1.** Schematic illustration of UCNPs@R-$TiO_2$ nanocomposite for photocatalytic sterilization under 980 nm NIR light irradiation.

## 2. Results and Discussion

### 2.1. Characterization of the UCNPs@R-TiO$_2$ Nanocomposites

The transmission electron microscopy (TEM) image shown in Figure 2A was recorded on a Tecnai G2 F20 microscope (USA) and was used to observe the crystal morphology

and size of the β-NaYF$_4$:Yb,Er,Gd fluorescent nanorods (UNCPs). The UNCPs were homo-dispersed and rod-shaped with a length of ~500 nm (Figure 2(Ba)) and a diameter of ~50 nm (Figure 2(Bb)). Moreover, the fast Fourier transform pattern indicated a (100) zone axis (Figure 2A). We found that the reduced TiO$_2$ (R-TiO$_2$) nanoparticles exhibited a square shape and were uniformly scattered (Figure 2C). The HRTEM (insert in Figure 2C) showed that R-TiO$_2$ belongs to pure anatase [11]. Additionally, the TEM image in Figure 2(Da) shows that the R-TiO$_2$ nanoparticles were successfully assembled on the UNCPs. Furthermore, we confirmed the crystal structure of the UCNPs@R-TiO$_2$ nanocomposite and found that the average lattice spacings that can be measured are 0.521 nm and 0.352 nm (Figure 2(Db)), matching well with a (100) facet and (101) facet lattice distance for the β-NaYF$_4$ and anatase TiO$_2$, respectively [11,21]. From energy-dispersive X-ray spectroscopy (EDS) measurements (shown in Figure S1), the elemental composition of the UNCPs@R-TiO$_2$ was obtained, and, as shown in the table of Figure 2(Dc), Na, Ti, Cu, Yb, F, Er, Gd, Y, and O could be detected; Cu originated from the Cu grid used for TEM measurements [26]. Taken together, these results clearly illustrate that the R-TiO$_2$ nanoparticles were success-fully assembled on the UCNPs. The FT-IR spectra of the UCNPs@R-TiO$_2$ nanocomposite was obtained on a Nexus 670 spectrophotometer and shown in Figure S2. A strong and broad absorption band at 464 cm$^{-1}$ was assigned to Ti–O and O–Ti–O flexion vibration originating from the TiO$_2$ crystals [27,28]. The wide band of around 3435 cm$^{-1}$ was at-tributed to the H–O stretching, which helped to enhance photocatalytic activity [27]. The band around 1557 cm$^{-1}$ was attributed to the carbonyl group (−C=O−) vibration [21]. The FT-IR spectrum analysis indicates that the R-TiO$_2$ nanoparticles were successfully assembled onto the surface of the as-synthesized UCNPs through electrostatic attraction.

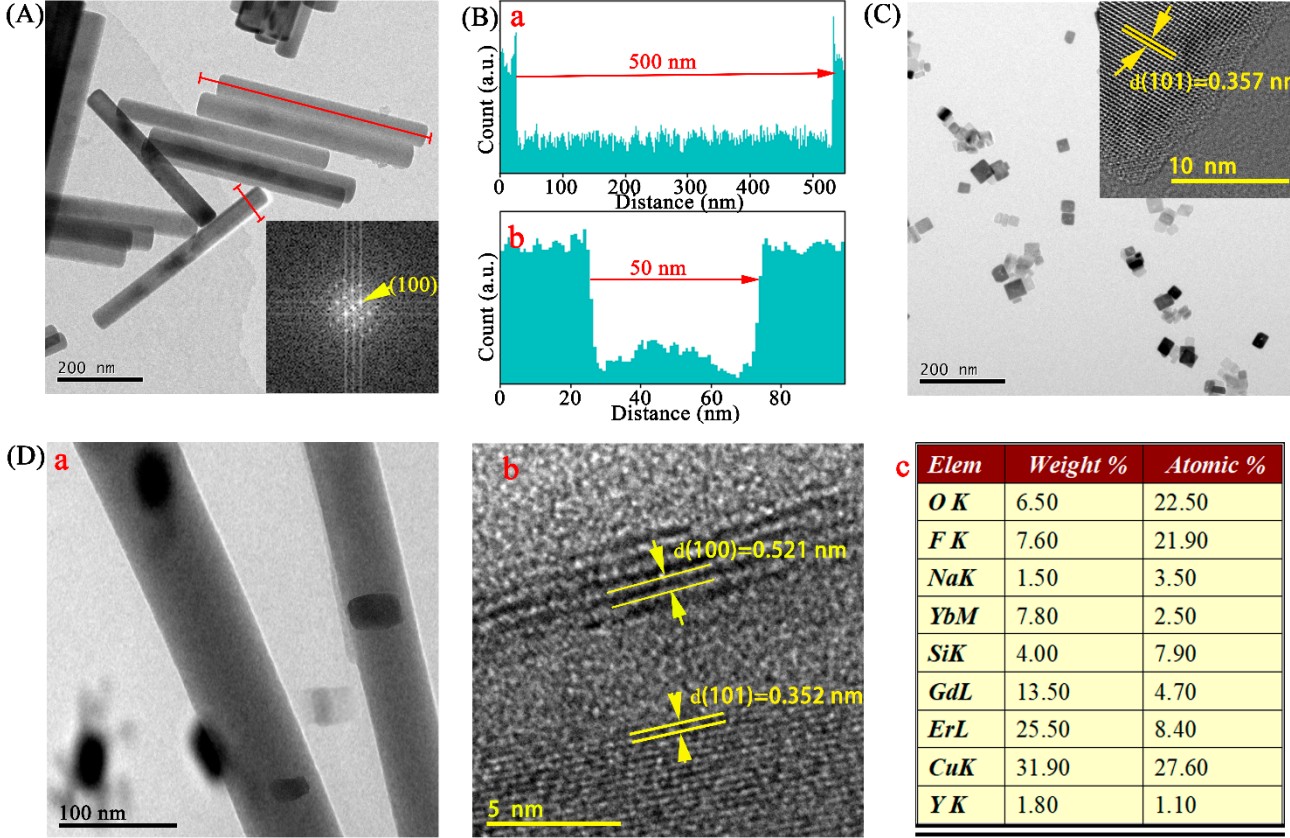

**Figure 2.** (**A**) TEM image of UCNPs, with insert showing the fast Fourier transform pattern. (**B**) Length (**a**) and diameter (**b**) analyses of (**A**). (**C**) TEM and HRTEM (high resolution TEM) images (inset) of R-TiO$_2$. (**D**) TEM image (**a**), HRTEM image (**b**), and elemental compositions analysis (**c**) of UCNPs@ R-TiO$_2$.

The XRD patterns were recorded on an X-ray diffractometer (D8 Advance, Brucker, Germany) with the Cu K radiation ($\lambda$ = 0.155 nm) operating at 40 kV and 80 mA; see Figure 3A and Figure S3. Figure S3 shows that the UCNPs and R-TiO$_2$ nanoparticles were pure hexagonal- (Joint Committee on Powder Diffraction Standards JCPDS 00-016-0334) and anatase-phase (JCPDS 01-021-1272), respectively. The sharp diffraction peaks indicate that the UCNPs and R-TiO$_2$ nanoparticles were highly crystallized hexagonal-and anatase-structured. The XRD pattern of the UCNPs@R-TiO$_2$ nanocomposite is shown in Figure 3A. And the XRD pattern analysis confirmed that the UCNPs@R-TiO$_2$ nanocomposite had a high degree of crystallization. The XRD pattern further showed that the UCNP nanorods were in a pure hexagonal phase (JCPDS 00-016-0334), which has been previously shown to have a higher luminous efficiency than the cubic phase NaYF$_4$ [29]. The distinct peaks at 25.3° of the prepared R-TiO$_2$ nanoparticles are likely ascribed to the (101) facet of the anatase TiO$_2$ when compared to the JCPDS 01-021-1272 database, indicating that the R-TiO$_2$ nanoparticles were present on the surface of the UCNP fluorescent nanorods by a form of substitutional doping. Compared to the peak locations of pure hexagonal and anatase phase, we observed that all of the diffraction peaks of the UCNPs@R-TiO$_2$ nanocomposite shifted to lower diffraction angles due to an expansion in unit-cell volume as a result of the partial substitution of Ti$^{4+}$ (65 pm) by the larger Y$^{3+}$ (104 pm) in the lattice [26].

As shown in Figure S4, the upconversion luminescence (UCL) intensity of the Gd-doped $\beta$-NaYF$_4$:Yb,Er (UCNPs) fluorescent nanorods recorded on a Hitachi F-7000 spectrometer was higher than that of the $\beta$-NaYF$_4$:Yb,Er fluorescent nanorods, which was attributed to the Gd dopant [21,30]. The UCL spectra of the UCNP nanorods and the UCNPs@R-TiO$_2$ nanocomposite were analyzed and shown in Figure 3(Ba,b). We found that, under 980 nm irradiation, the UCNP nanorods emit intense UCL emissions at 523, 542, and 658 nm, which were assigned to the $^2H_{11/2}$–$^4I_{15/2}$, $^4S_{3/2}$–$^4I_{15/2}$, and $^4F_{9/2}$–$^4I_{15/2}$ transitions of Er$^{3+}$ (Figure 1), respectively [21]. By contrast, the UCNPs@R-TiO$_2$ exhibited a drastic reduction to the UCL intensity because of the energy transfer from UCNPs to R-TiO$_2$ [20]. The absorption spectrum in the UV-Vis range (Figure 3C) was recorded on a UV-visible Cary 300 spectrophotometer and indicated that the R-TiO$_2$ (Figure 3(Ca)) possesses a higher absorption in the visible light region than pure anatase TiO$_2$ (Figure 3(Cb)), which was caused by the oxygen vacancies and lower bandgap of the R-TiO$_2$ (2.8 eV) than that of TiO$_2$ (3.2 eV) (inset in Figure 3C and Equation (S1)) [10]. The oxygen vacancies and low bandgap of the R-TiO$_2$, which arose from the dopant of Ti$^{3+}$ under an argon atmosphere, are helpful to enhance the absorption of visible light [10,11]. In addition, we observed that the emission peaks of UCNPs can match the enhanced visible light absorption of the R-TiO$_2$ nanoparticles (see the dotted boxes in Figure 3C), based on the UV-Vis absorption spectrum data (Figure 3C). We also found that the zeta potentials of the UCNP nanorods (Figure 3(Da)) and R-TiO$_2$ nanoparticles (Figure 3(Db)) were negative and positive, respectively, indicating that R-TiO$_2$ can be coupled to the surface of UCNP nanorods in a solution by electrostatic attraction. As a result, the UCNPs@R-TiO$_2$ composites possessed a positive zeta potential (Figure 3(Dc)) that helps its binding to the negatively charged surface of *E.coli* bacteria [31].

### 2.2. Antibacterial Performance

The plate-counting bacteria colonies of *E. coil* were used to evaluate the bacteriostatic ability of UCNPs@R-TiO$_2$ composites under 980 nm light irradiation (1 W). Under the different preparation conditions (Table S1), the highest antibacterial efficiency (98.1%) was achieved for the UCNPs@R-TiO$_2$ (180 °C, 20 h) composites. The low reaction temperature (180 °C) was unfavorable to the crystal growth of R-TiO$_2$, and the high reaction temperature (300 °C) caused the agglomeration of R-TiO$_2$, which was not good for photocatalytic reactions [11,31]. Simultaneously, the high crystallinity of R-TiO$_2$ was achieved at the optimal reaction time (20 h) [11]. The highest antibacterial efficiency (98.7%) was also achieved on the UCNPs@R-TiO$_2$ (40%) composites among the different mass ratios of R-TiO$_2$ and UCNPs (Figure S5). Consequently, the optimal UCNPs@R-TiO$_2$ (30%, 180 °C, 20 h) composites

were used for further research. As shown in Figure 4A, the bacterial photoinactivation effect was suitably correlated with the dosage of nanomaterials and the UCNPs@R-TiO$_2$ nanocomposite, resulting in the highest bactericidal effect (97.3%) at the concentration of 50 μg/mL. Interestingly, we further found that both the UCNPs (Figure 4(Ba)) and R-TiO$_2$ (Figure 4(Bb)) were capable of killing *E. coil* colonies on the agar plate under 980 nm laser irradiation for 12 min when compared to the saline control (Figure 4B). Strikingly, the UCNPs@R-TiO$_2$ nanocomposite (Figure 4(Bc) and Figure S6) treatment eliminated about 98.1% of *E. coil* colonies on the plate, reflecting its enhanced bactericidal activity. Simultaneously, the bactericidal performance of the optimized UCNPs@R-TiO$_2$ nanocomposite was compared to the previously reported works (Table S2). We also found that the UCNPs@R-TiO$_2$ nanocomposite had the best bactericidal effect among these antibacterial agents.

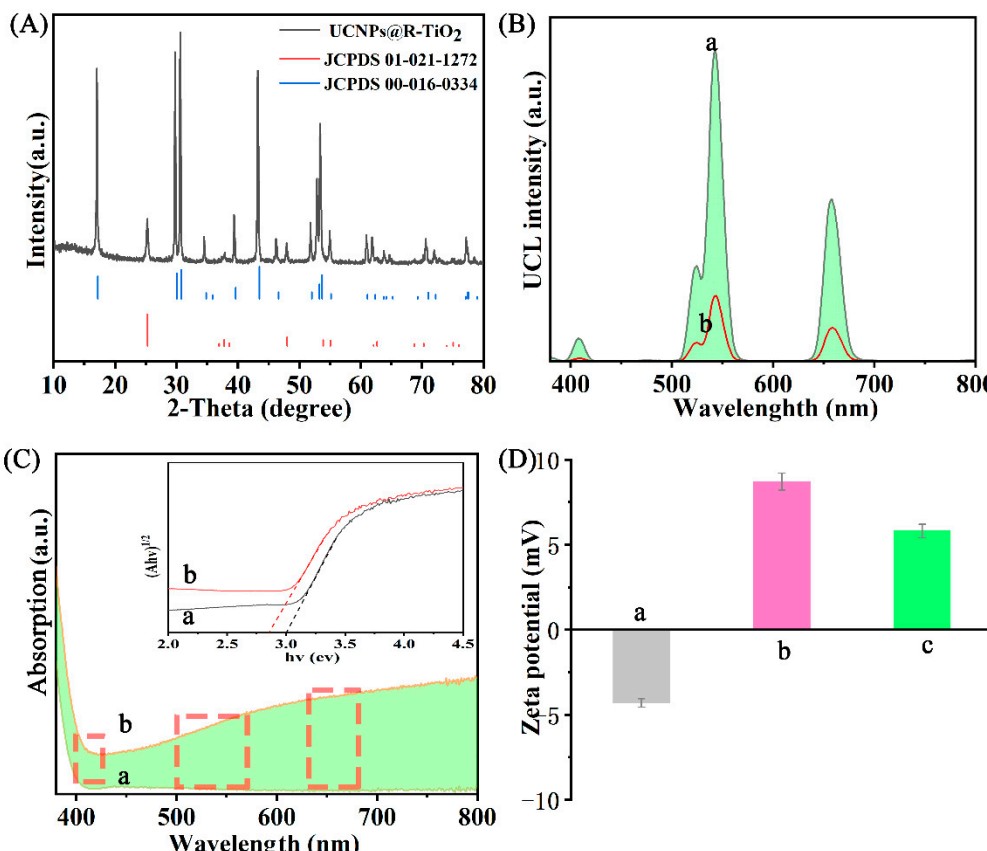

**Figure 3.** (**A**) X-ray powder diffraction (XRD) patterns of UCNPs@R-TiO$_2$ and the standard hexagonal phase (JCPDS 00-016-0334) and anatase phase (JCPDS 01-021-1272). (**B**) The upconversion luminescence (UCL) spectrum of the UCNPs (a) and UCNPs@R-TiO$_2$ (b). (**C**) The UV-Vis absorption spectrum of TiO$_2$ (a) and R-TiO$_2$ (b), with insert showing the corresponding bandgap determined by Tauc plot. (**D**) The zeta potential of UCNPs (a), R-TiO$_2$ (b), and UCNPs@R-TiO$_2$ (c).

The sterilization of these three materials under the same condition was also investigated via matrix-assisted laser desorption/ionization time-of-flight mass spectrometry (MALDI-TOF MS) analysis. As shown in Figure 5a, there were distinct characteristic peaks ($m/z$ = 4331, 5060, 6220, 7250, 9190, 9519) of *E. coli* K12 [32–34] present in our analysis, indicating the massive survival of *E. coli* K12. However, the number of peaks was dramatically decreased upon adding either the UCNP nanorods (Figure 5b) or R-TiO$_2$ nanoparticles (Figure 5c), indicating that some of the bacteria were killed. Importantly, there were no characteristic peaks detected between the 4000 to 14,000 Da region after the treatment of the UCNPs@R-TiO$_2$ composites, indicating that the *E. coli* K12 was nearly entirely killed.

All these data demonstrate that the UCNPs@R-TiO$_2$ nanocomposites possessed a highly effective bactericidal ability under the 980 nm NIR illumination.

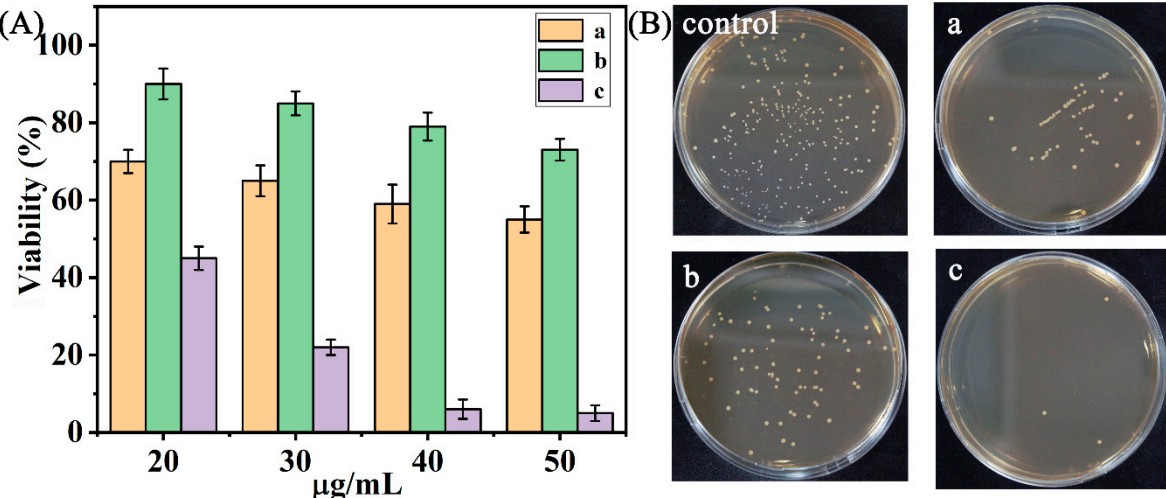

**Figure 4.** (**A**) *E. coli* viability under all different sample concentrations and (**B**) photographs of agar plates of *E. coli* incubated with 40 µg/mL of UCNPs (**a**), R-TiO$_2$ (**b**), and UCNPs@R-TiO$_2$ nanocomposite (**c**) using a 980 nm laser (1 W, 12 min).

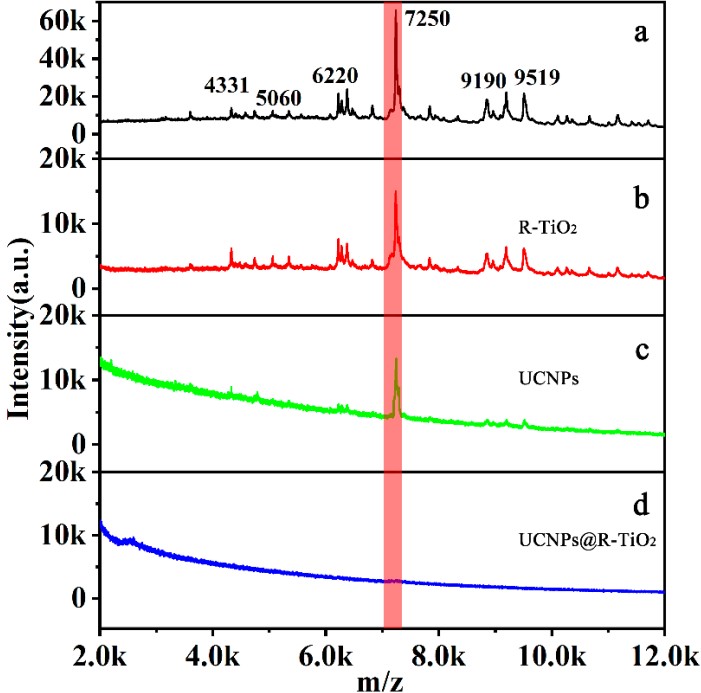

**Figure 5.** MALDI-TOF MS analysis of *E. coli* without (**a**) and with R-TiO$_2$ (**b**), UCNPs (**c**), and UCNPs@R-TiO$_2$ composites (**d**) under 980 nm NIR light irradiation for 20 min (40 µg/mL). The red band is the characteristic peak of *E. coli* at *m/z* = 7250.

### 2.3. Cytotoxicity Assessment

We evaluated the potential cytotoxicity to cells (HEK 293) of the as-prepared nanocomposites via MTT assay. There was about 80%, 87%, and 98% viability of HEK293 cells upon the treatment of the UCNPs, R-TiO$_2$, and UCNPs@R-TiO$_2$ materials at a concentration of 12 µg/mL, indicating low cytotoxicity to mammalian cells (Figure 6). It was noted that the cell viability was largely decreased to around 68% when the UCNP concentration

was up to 50 μg/mL. By contrast, R-TiO$_2$ showed almost no toxicity to HEK 293 cells, because the HEK 293 cells still manifested a high survival rate (more than 91%) even under a high concentration (100 μg/mL) condition. Notably, the cell viability of the UCNPs@R-TiO$_2$ treatment was improved, possibly due to the low toxicity of R-TiO$_2$. About 80% of cells survived under a high concentration (40 μg/mL) of UCNPs@R-TiO$_2$ incubation, demonstrating its potential for bactericide material application [35].

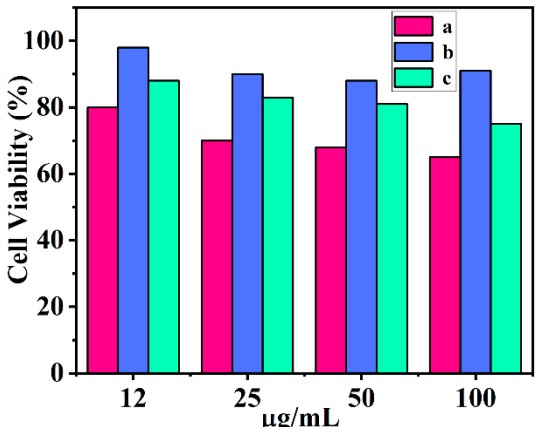

**Figure 6.** Cytotoxicity of UCNPs (**a**), R-TiO$_2$ (**b**), and UCNPs@R-TiO$_2$ nanocomposite (**c**) in HEK 293 cells.

### 2.4. Antibacterial Mechanism

Figure 7 depicts the possible 980 nm NIR light-driven antibacterial mechanism of the UCNPs@R-TiO$_2$ nanocomposites. After the 980 nm NIR light irradiation, the $^2F_{7/2}$ state electrons of Yb$^{3+}$ would be promoted into the $^2F_{5/2}$ excited state band of Yb$^{3+}$, then the Yb$^{3+}$ $^2F_{5/2}$ state would be relaxed by energy transfer to a neighboring Er$^{3+}$ ion. The energy promotes the valence band ($^4I_{15/2}$) electrons of Er$^{3+}$ into the excited state band ($^2H_{11/2}$, $^4S_{3/2}$, or $^4F_{9/2}$). The electrons in the states H, S, or F in Er$^{3+}$ are unstable and would be relaxed to the $^4I_{15/2}$ state by releasing energy, emitting at 523 nm, 542 nm, and 658 nm, respectively [21,35,36]. The visible light energy would be further absorbed by the valence band electrons of neighboring R-TiO$_2$ nanoparticles. The high energy valence electrons would jump into the stable conduction band, producing electron-hole pairs.

The visible light (red and green light) emitted by UCNPs is absorbed by the R-TiO$_2$ to produce strongly reductive electrons (e$^-$) and oxidative holes (h$^+$). The valid h$^+$/e$^-$ pairs afterwards could react with H$_2$O and O$_2$ in an aqueous solution (Equations (1)–(7)) to produce reactive species. As it is known, the generated hydroxyl radical (·OH) species can be used as a strong oxidizer for the non-selective killing of bacteria [37]. The amount of ·OH is detected by the fluorescent intensity of 2-hydroxyterephthalic acid ($\lambda$ = 420 nm) which is a product of the reaction of ·OH with terephthalic acid [38]. As shown in Figure 7, the UCNPs@R-TiO$_2$ nanocomposite has the highest fluorescent intensity among these materials, indicating the amount of OH that was generated. These results show that the visible light emitted by UCNPs can be effectively absorbed by R-TiO$_2$ nanoparticles. The UCNPs@R-TiO$_2$ composites possess excellent photocatalytic performance and the specific process of the NIR photocatalytic sterilization of *E. coli* is summarized using the following reactions:

$$\text{UNCPs} + \text{NIR light} \rightarrow \text{Visible light,} \tag{1}$$

$$\text{R-TiO}_2 + \text{Visible light} \rightarrow \text{h}^+ + \text{e}^- \tag{2}$$

$$\text{O}_2 + \text{e}^- \rightarrow \cdot\text{O}_2{}^-, \tag{3}$$

$$2\cdot\text{O}_2{}^- + 2\text{H}^+ \rightarrow \text{H}_2\text{O}_2 + \text{O}_2, \tag{4}$$

$$\cdot O_2{}^- + H_2O_2 \rightarrow \cdot OH + OH^- + O_2, \tag{5}$$

$$H_2O + h^+ \rightarrow \cdot OH + H^+, \tag{6}$$

$$\cdot OH + E.\ coli \rightarrow \text{Inactivated } E.\ coli. \tag{7}$$

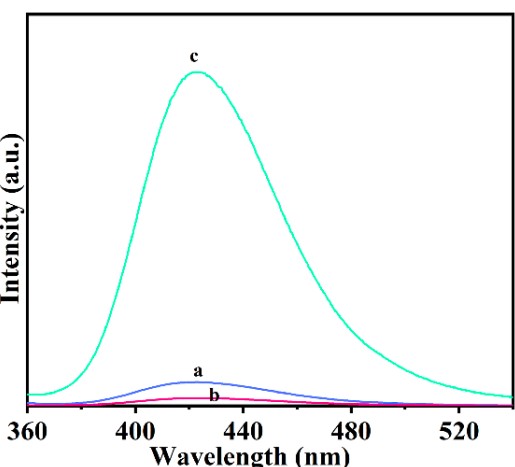

**Figure 7.** Photoluminescence spectra of UCNPs (**a**), R-TiO$_2$ (**b**), and UCNPs@R-TiO$_2$ nanocomposite (**c**), respectively, measured under 980 nm NIR illumination for 20 min.

## 3. Experimental Designs

### 3.1. Reagents and Materials

All of the chemical reagents—at analytical grade, unless otherwise noted—were used without further purification. Ytterbium(III) chloride hexahydrate (YbCl$_3$·6H$_2$O), Yttrium(III) chloride hexahydrate (YCl$_3$·6H$_2$O), Gadolinium(III) chloride (GdCl$_3$·6H$_2$O), and Erbium(III) chloride (ErCl$_3$) were purchased from Alfa Aesar (Shanghai, China). Titanium isopropoxide, sodium hydroxide (NaOH, 96%), oleic acid (OA), sodium citrate, hydrofluoric acid, acetonitrile, ethanol, and chloroform of analytical grade were purchased from Sinopharm Chemical Reagent Co., Ltd. from Shanghai, China. NH$_4$F and Thiazolyl Blue Tetrazolium Bromide (MTT) were obtained from Aladdin (Hang Kong, China) and Bomei Biotechnology (Hefei, China), respectively. Ultrapure water (18.2 MΩ·cm, Mili-Q, Millipore, Burlington, MA, USA) was used throughout the experiment.

### 3.2. Preparation of β-NaYF$_4$:Yb,Er,Gd (60, 18, 2, 20 mol%) Fluorescent Nanorods

The preparation of β-NaYF$_4$:Yb,Er,Gd (60, 18, 2, 20 mol%) highly fluorescent nanorods (UCNPs) is summarized as follows [21,39]: Firstly, a mixed solution with 10 mL of ethanol, 3 mL of deionized water, and 0.6 g of NaOH was obtained. A total of 10 mL of oleic acid was then dropped in the above mixture under vigorous stirring. The stirring was continued for 20 min; the obtained mixture was named A solution. Moreover, 145.6 mg of YCl$_3$·6H$_2$O, 59.47 mg of GdCl$_3$·6H$_2$O, 4.38 mg of ErCl$_3$, and 55.8 mg of YbCl$_3$·6H$_2$O were dissolved in 4 mL of deionized water; this mixture was named B solution. Then, B solution was slowly dropped into the A solution under vigorous stirring. After 10 min, 2.0 mL of NH$_4$F (2 M) solution was added to the above mixture dropwise. Finally, a milky colloidal solution was produced. The obtained solution was encapsulated in a 50 mL autoclave flask and heated to 180 °C at a heating rate of 3 °C/min, then it was kept at 200 °C for 2 h. The prepared UCNPs were collected and washed with ethanol and ultrapure water, respectively.

### 3.3. Synthesis of UCNPs@R-TiO$_2$ Nanocomposite

UCNPs@R-TiO$_2$ were prepared as follows [11,20,40]. Firstly, the R-TiO$_2$ nanoparticles (TiO$_{2-x}$) were prepared as follows: 4 μL of titanium isopropoxide was dropped into a solution containing 1.2 mL of HF and 20 mL of isopropanol. After the obtained mixture

was stirred for 10 min under an argon atmosphere, 5 μL of TiCl$_3$ solution was added to it [11]. Then, 20 mg of UCNPs were also added to the obtained mixture under an ultrasonic bath and the mixture solution continued to sonicate for 30 min under an argon atmosphere. The obtained mixture was then encapsulated into a 50 mL autoclave flask, heated to 180 °C at a heating rate of 3 °C/min, and kept at 180 °C for 20 h under an argon atmosphere. The produced UCNPs@R-TiO$_2$ (40%) nanocomposite was collected and washed, first with ethanol and then with ultrapure water. It was then dried at 60 °C for 10 h in a vacuum environment. The R-TiO$_2$ nanoparticles and UCNPs@R-TiO$_2$ nanocomposites with different mass ratios were prepared according to the above method by changing the amount of UCNPs.

### 3.4. Vitro Cell Viability Assay

A standard MTT assay was used to assess the cytotoxicity of the UCNPs, R-TiO$_2$, and UCNPs@R-TiO$_2$. Human Embryonic Kidney 293 cells (HEK293) were selected for the assay. The temperature of the whole incubation process was controlled at 37 °C. The HEK293 cells present in a 96-well plate (with 10,000 cells per well) were first cultured for 12 h. After that, the different calculated concentrations of UCNPs, R-TiO$_2$, and UCNPs@R-TiO$_2$ nanomaterials were added to the above plate and the mixture was further incubated for 24 h. Afterwards, 100 μL of MTT solution was added to each well. After incubation for 2 h, the sediment was retained. Then, 100 μL of DMSO (dimethyl sulfoxide) was added into each sample mentioned above and the mixture was shaken for 20 min. The absorbance at 595 nm detected by a microplate reader was used to calculate the cell viability rate.

### 3.5. Bacteria (E. coli K12) Culture and Preparation

*E. coli* K12 were inoculated into the Luria-Bertani broth and shaken (300 rpm) constantly in an incubator shaker at 37 °C. After overnight incubation, the *E. coli* K12 suspensions (8 mL) were centrifuged (8000 rpm) for 2 min and the sediments were retained. The obtained precipitates were resuspended by adding 6 mL of sterile normal saline. By measuring the optical density (OD) value at 600 nm, the bacterium liquid concentration was adjusted to a proper level for after use. Different colonies were distributed on LB plates and incubated at 37 °C overnight. The relevant colony-forming units (CFU) were calculated to obtain the number of bacteria per milliliter.

### 3.6. Antibacterial Properties

The Gram-negative bacterium *E. coli* K-12 was used to investigate the in vitro antibacterial abilities of the UCNPs, R-TiO$_2$, and UCNPs@R-TiO$_2$. An amount of 900 μL of *E. coli* suspension (~10$^6$ CFU/mL) and 100 μL of these nanomaterials at different concentrations were mixed. The resulting concentrations were 20, 30, 40, and 50 μg/mL, respectively. After incubation at 37 °C for 2 h, the obtained bacterial suspension was diluted by a factor of 10$^3$. The resulting bacterial samples were irradiated for 20 min by a 980 nm NIR light (1 W). Afterwards, 100 μL of the above suspension was spread on the Luria-Bertani medium and hatched at 37 °C for 16 h. In the end, the antibacterial abilities were assessed on the LB agar plates using the colony counting method. Simultaneously, instead of the added nanocomposites, an isotonic saline solution was added into the *E. coli* suspension as a blank control.

### 3.7. MALDI-TOF MS Analysis

The characterization changes of *E. coli* K12 bacterial strains were analyzed using a MALDI-TOF MS [11,41–43]. Firstly, 300 μL of ultrapure water and 900 μL of ethanol were mixed, and then 20 mg of the *E. coli* sample was added into the mixture under mild shaking. The sediment was retained after centrifuging (13,000 rpm) for 3 min. Subsequently, 50 μL of CH$_3$CN (acetonitrile) and 50 μL of 70% HCOOH (formic acid) were added. The mixture was centrifuged at 13,000 rpm for 2 min again. The above mixture (0.5 μL) and a DHB matrix solution (2,5-dihydroxybenzoic acid solution, 0.5 μL) were dropped on a

plate and allowed to dry. The experiments of the MALDI-TOF MS were performed on an UltrafleXtreme TOF/TOF operating system equipped with a 355 nm $N_2$ laser. The operating conditions were as follows: positive ion mode, mass range (5–20 kDa), and acceleration voltage (20 kV).

## 4. Conclusions

The UCNPs@R-TiO$_2$ nanocomposite, an effective antibacterial material, was prepared with the electrostatic assembly strategy. The reduced TiO$_2$ nanoparticles with a bandgap of 2.8 eV and the ability to absorb visible light were successfully assembled onto the surface of Gd-enhanced β-NaYF$_4$:Yb,Er fluorescent nanorods. The UCNPs@R-TiO$_2$ composite with an MIC of 40 μg/mL can kill more than 98.1% of *E. coli* within 12 min under 980 nm NIR light irradiation (1 W). The good antibacterial properties are mostly attributed to the efficient light energy transfer from UCNPs to R-TiO$_2$ nanoparticles. The low-toxicicity UCNPs@R-TiO$_2$ nanocomposite shows great potential for creating efficient NIR-responsive photocatalysts.

**Supplementary Materials:** The following are available online at https://www.mdpi.com/2073-4344/11/2/184/s1: Figures S1–S6: EDX, FT-IR, XRD, UCL and Antibacterial efficiency analysis of the prepared nanomaterials, Table S1: Antibacterial efficiency of UCNPs@R-TiO$_2$ nanocomposite under different preparation conditions, Table S2: Comparison of the performance of UCNPs@R-TiO$_2$ photocatalytic sterilization system with that of some antibacterial agents.

**Author Contributions:** H.Z.: the acquisition and analysis of data for the work; Drafting the work; Final approval of the version to be published; Agreement to be accountable for all aspects of the work. F.H.: the conception or design of the work; revising the work; Final approval of the version to be published; Agreement to be accountable for all aspects of the work. All authors have read and agreed to the published version of the manuscript.

**Funding:** This research funded by National Natural Science Foundation of China, grant number 21275042.

**Institutional Review Board Statement:** The study was conducted and approved by the Ethics Committee of University of South China (SYXK(湘)2020-0002 and January, 2020).

**Informed Consent Statement:** Informed consent was obtained from all subjects involved in the study.

**Data Availability Statement:** Data sharing not applicable.

**Conflicts of Interest:** There are no conflict of interest to declare.

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
