# Peer review of "Using Gd-Enhanced β-NaYF4:Yb,Er Fluorescent Nanorods Coupled to Reduced TiO2 for the NIR-Triggered Photocatalytic Inactivation of Escherichia coli"

_catalysts, doi:10.3390/catal11020184_

Round 1

Reviewer 1 Report

In the manuscript entitled Gd enhanced β-NaYF4:Yb,Er fluorescent nanorods coupled reduced TiO2 for NIR-triggered photocatalytic inactivation of Escherichia coli the authors describe the synthesis and characterization of the β-NaYF4:Yb,Er,Gd nanorods, reduced TiO2 nanoparticles and a composite formed by these materials joined by electrostatic forces in different molar ratios. Besides, the authors have shown the photoactivity of these materials as antibacterial behavior for Escherichia coli. The manuscript is scientifically and technically sound. The text reads quite well and deals with a very interesting topic. The introduction is well documented, and the description of the results is rather clear. The authors also show a very complete analysis of the great number of results. However, the authors have to account for some suggestions before the publication of the manuscript in the Catalysts Journal.

Main results:

The authors don´t mention some important information about the material characterization such as Transmission Electron Microscopy, Photoluminescence, X-Ray Diffraction measurements, Fourier-Transform Infrared Spectroscopy. A brief description of the characterization conditions must be included.

In section 3.1. Characterization of the UCNPs@R-TiO2 Nanocomposites the authors mention the characterization of the crystal morphology and granular size of the β-NaYF4:Yb,Er,Gd fluorescent nanorods (UNCPs). It is measured the size of the nanorods but not the grain size since it is not clear if they are monocrystalline.

In the first paragraph after Figure 2, the authors mention a UV-vis spectrum. It is more appropriate to write an absorption spectrum in the UV-vis range.

In section 3.2. Antibacterial Performance. The authors mention the high crystallinity of R-TiO2 was achieved at the optimal reaction time (20 h). The authors don´t show the evolution of the crystallinity as a function of the temperature. Where the results come from? On the other hand, the higher antibacterial response shown in Figure S5 seems to be for a molar ratio of 40% instead of 30%. In addition, in Figure 4A it is shown the bacterial photoinactivation effect is higher for 50 μg/mL instead of 40 μg/mL. The text and the Figure seem to have no correlation. Could be it described in a better manner?

Section Cytotoxicity Assessment has to be rephrased. The authors describe the figure in a vague manner. For example the author mention: There were about 80% of viability of HEK293 cells upon three materials treatment at a concentration of 12 µg/mL. In the figure, the values range from 80 to close to 100% for a concentration of 12 µg/mL. This doesn´t mean viability of about 80% for all the materials. Please, revise the whole paragraph, there are some inconsistencies.

In the paragraph after Figure 7 the authors mention: The amount of ·OH is detected by the fluorescent intensity of 2-hydroxyterephthalic acid which is a reaction of ·OH and terephthalic acid. Could be possible to include in the text the wavelengths in which they appear?

Finally, the manuscript showed very good results but the conclusions are very brief. The manuscript could be improved if a better description of the results is shown in the conclussions.

Figures:

In general, the figures are clear and well represented. However, the figure captions could be improved, including a better description of the figure. In some cases, it is not clear the presented results, and also there are some mistakes.

For example, in Figure 3, the technique used to characterize the materials' crystal structure is the X-Ray powder Diffraction (XRD) method. Also, in this figure are shown the reference diffraction patterns for TiO2 and UCNPs. Also, a brief description of the squares shown in Figure 3C could improve the understanding of the figure (even when mentioned in the text). It is understood, the authors have measured the optical density or absorption of the TiO2 nanoparticles. In this sense, the y-axis of figure 3C uses to be called Absorption. UV-Vis Intensity seems to be not so clear.

Figure 4 A. The y-axis is not well labeled: vialbility instead of viability.

In figure 5, the red band used to emphasize the peak at m/z=7250 is not explained in the figure caption. Also in this figure, there is no correspondence between the figure caption and the description in the text. Please revise the correctness of them. The materials' names in the figure could improve the understanding of the whole figure.

English revision:

  • Abstract: the word absorption instead of adsorption in line 4 and also in the introduction in the second paragraph in the second line. Please revise the text if there are more mistakes.
  • In the last line before Figure 2, probably the authors want to use the word larger instead of lager.
  • In section 2.4. Vitro Cell Viability Assay. change of the unities 10000 cells per cell.
  • Revision of the symbol for the Celsius Degrees.

Author Response

Comments and Suggestions for Authors: In the manuscript entitled Gd enhanced β-NaYF4:Yb,Er fluorescent nanorods coupled reduced TiO2 for NIR-triggered photocatalytic inactivation of Escherichia coli the authors describe the synthesis and characterization of the β-NaYF4:Yb,Er,Gd nanorods, reduced TiO2 nanoparticles and a composite formed by these materials joined by electrostatic forces in different molar ratios. Besides, the authors have shown the photoactivity of these materials as antibacterial behavior for Escherichia coli. The manuscript is scientifically and technically sound. The text reads quite well and deals with a very interesting topic. The introduction is well documented, and the description of the results is rather clear. The authors also show a very complete analysis of the great number of results. However, the authors have to account for some suggestions before the publication of the manuscript in the Catalysts Journal.

(1) The authors don´t mention some important information about the material characterization such as Transmission Electron Microscopy, Photoluminescence, X-Ray Diffraction measurements, Fourier-Transform Infrared Spectroscopy. A brief description of the characterization conditions must be included.

Reply: Thank you very much for your suggestion. The brief descriptions of the characterization conditions have been added into the paper. Please see pages 5 and 6 of the revised manuscript.

(2) In section 3.1. Characterization of the UCNPs@R-TiO2 Nanocomposites the authors mention the characterization of the crystal morphology and granular size of the β-NaYF4:Yb,Er,Gd fluorescent nanorods (UNCPs). It is measured the size of the nanorods but not the grain size since it is not clear if they are monocrystalline.

Reply: Thank you very much for your suggestion. The clear description has been revised in the revised manuscript. Please see page 5, line 158.

(3) In the first paragraph after Figure 2, the authors mention a UV-vis spectrum. It is more appropriate to write an absorption spectrum in the UV-vis range.

Reply: Thank you very much for your suggestion. The related expression has been revised in the manuscript. Please see page 6, line 213.

(4) In section 3.2. Antibacterial Performance. The authors mention the high crystallinity of R-TiO2 was achieved at the optimal reaction time (20 h). The authors don´t show the evolution of the crystallinity as a function of the temperature. Where the results come from? On the other hand, the higher antibacterial response shown in Figure S5 seems to be for a molar ratio of 40% instead of 30%. In addition, in Figure 4A it is shown the bacterial photoinactivation effect is higher for 50 μg/mL instead of 40 μg/mL. The text and the Figure seem to have no correlation. Could be it described in a better manner?

Reply: Thank you very much for your suggestions. The section 3.2 has been revised in the manuscript. Please see page 7, line 235-250.

(5) Section Cytotoxicity Assessment has to be rephrased. The authors describe the figure in a vague manner. For example the author mention: There were about 80% of viability of HEK293 cells upon three materials treatment at a concentration of 12 µg/mL. In the figure, the values range from 80 to close to 100% for a concentration of 12 µg/mL. This doesn´t mean viability of about 80% for all the materials. Please, revise the whole paragraph, there are some inconsistencies.

Reply: Thank you very much for your suggestion. The paragraph has been rephrased in the revised manuscript. Please see page 9, line 276-278.

(6) In the paragraph after Figure 7 the authors mention: The amount of ·OH is detected by the fluorescent intensity of 2-hydroxyterephthalic acid which is a reaction of ·OH and terephthalic acid. Could be possible to include in the text the wavelengths in which they appear?

Reply: Thank you very much for your suggestion. The wavelength has been added into the revised manuscript. Please see page 10, line 309.

(7) Finally, the manuscript showed very good results but the conclusions are very brief. The manuscript could be improved if a better description of the results is shown in the conclusions.

Reply: Thank you very much for your suggestion. The better description of the results has been added in the conclusions of manuscript. Please see page 11, line 317-320.

(8, 9, 10) In general, the figures are clear and well represented. However, the figure captions could be improved, including a better description of the figure. In some cases, it is not clear the presented results, and also there are some mistakes.

For example, in Figure 3, the technique used to characterize the materials' crystal structure is the X-Ray powder Diffraction (XRD) method. Also, in this figure are shown the reference diffraction patterns for TiO2 and UCNPs. Also, a brief description of the squares shown in Figure 3C could improve the understanding of the figure (even when mentioned in the text). It is understood, the authors have measured the optical density or absorption of the TiO2 nanoparticles. In this sense, the y-axis of figure 3C uses to be called Absorption. UV-Vis Intensity seems to be not so clear. Figure 4 A. The y-axis is not well labeled: vialbility instead of viability.

In figure 5, the red band used to emphasize the peak at m/z=7250 is not explained in the figure caption. Also in this figure, there is no correspondence between the figure caption and the description in the text. Please revise the correctness of them. The materials' names in the figure could improve the understanding of the whole figure.

Reply: Thank you very much for your suggestions. The figures have been improved in the revised manuscript. Please see Figures 3, 4 and 5, lines 229-230, 273-274.

(11) English revision:

Abstract: the word absorption instead of adsorption in line 4 and also in the introduction in the second paragraph in the second line. Please revise the text if there are more mistakes.

In the last line before Figure 2, probably the authors want to use the word larger instead of lager.

In section 2.4. Vitro Cell Viability Assay. change of the unities 10000 cells per cell.

Revision of the symbol for the Celsius Degrees.

Reply: Thank you very much for your suggestions. The relevant written mistakes have been revised in the whole manuscript. Please see the revised manuscript.

Reviewer 2 Report

The article present interesting results to the redearship of the journal. However, before publication, the article needs extensive editing of English. I upload a file with some corrections I have tried to make in order to improve the article in terms of comprehending its content.

Author Response

Responses to Reviewer 2:

Comments and Suggestions for Authors: The article present interesting results to the redearship of the journal. However, before publication, the article needs extensive editing of English. I upload a file with some corrections I have tried to make in order to improve the article in terms of comprehending its content.

Thank you very much for your comments and suggestions. The whole article has been edited extensively according to your suggestions. Please see the revised manuscript.

Reviewer 3 Report

Evaluation

Gd enhanced β-NaYF4:Yb,Er fluorescent nanorods coupled

reduced TiO2 for NIR-triggered photocatalytic inactivation of

Escherichia coli

Comments:

The aim of this paper was to kill Escherichia coli in a short time by NIR light irradiation at a MIC of UCNPs@R-TiO2 nanocomposite. β-NaYF4:Yb,Er,Gd fluorescent nanorods coupled reduced TiO2 (UCNPs@R-TiO2) nanocomposite was successfully prepared and applied to visible-light catalytic sterilization under a NIR light illumination.

In this manuscript, only few self-citations can be found, and after checking them thoroughly, we can find out that all of them can be considered relevant in this research.

Zhou et al. after preparing the above mentioned effective antibacterial material by the electrostatic assembly strategy, the reduced TiO2 nanoparticles were successfully assembled onto the surface of Gd enhanced β-NaYF4:Yb,Er fluorescent nanorods. For the characterization and analysis of UCNPs@R-TiO2 XRD, TEM, UCL, UV-VIS and elemental analyses were applied, which provide a certain proof of the structure.

This manuscript contains mainly usable results and enough novelty in this field, article needs some improvements. Therefore, I suggest the minor revision of this manuscript.

Suggested revisions:

  1. Sometimes the names of journals are abbreviated, sometimes not. All of them should be abbreviated. For example, in citation 18, Nature materials should be changed to Nat. Mater.
  2. On Figure 2., the elemental composition analysis of UCNPs@R-TiO2 the weight % and atomic % should be corrected for example 06.50 to 6.50, or 01.50 to 1.50, etc..
  3. On Figure 3, wavelenhgth should be corrected to wavelength
  4. On page 4. acetonitrile should be written as CH3CN instead C2H3N illustrating better the real structure of the molecule
  5. On page 6, line 200, figure should be corrected to Figure.
  6. Sometimes space is added between the value and %, sometimes not. Authors should not write it. For example p. 7. 230, 236 or 237.
  7. Please improve the quality of EDX image in Figure 3, on the insert showing part of the corresponding bandgap determined by Tauc plot (UV-vis absorption spectrum)
  8. On page 7, coil should be changed to E. coli (three times). On p. 10, line 306. E. coli should be written with italics.

All things considered, this manuscript is a well-written text and showed a valuable preparation method with possible future photocatalytic application in visible-light catalytic sterilization. Therefore, after minor revision, the manuscript can be published in this journal.

Author Response

Responses to Reviewer 3:

Comments: The aim of this paper was to kill Escherichia coli in a short time by NIR light irradiation at a MIC of UCNPs@R-TiO2 nanocomposite. β-NaYF4:Yb,Er,Gd fluorescent nanorods coupled reduced TiO2 (UCNPs@R-TiO2) nanocomposite was successfully prepared and applied to visible-light catalytic sterilization under a NIR light illumination.

In this manuscript, only few self-citations can be found, and after checking them thoroughly, we can find out that all of them can be considered relevant in this research.

Zhou et al. after preparing the above mentioned effective antibacterial material by the electrostatic assembly strategy, the reduced TiO2 nanoparticles were successfully assembled onto the surface of Gd enhanced β-NaYF4:Yb,Er fluorescent nanorods. For the characterization and analysis of UCNPs@R-TiO2 XRD, TEM, UCL, UV-VIS and elemental analyses were applied, which provide a certain proof of the structure.

This manuscript contains mainly usable results and enough novelty in this field, article needs some improvements. Therefore, I suggest the minor revision of this manuscript.

(1) Sometimes the names of journals are abbreviated, sometimes not. All of them should be abbreviated. For example, in citation 18, Nature materials should be changed to Nat. Mater.

Reply: Thank you very much for your suggestion. The names of journals have been abbreviated in the revised manuscript. Please see the references section.

(2) On Figure 2., the elemental composition analysis of UCNPs@R-TiO2 the weight % and atomic % should be corrected for example 06.50 to 6.50, or 01.50 to 1.50, etc..

Reply: Thank you very much for your suggestion. The expression of elemental composition analysis has been revised in the manuscript. Please see page 6, Figure 2.

(3) On Figure 3, wavelenhgth should be corrected to wavelength.

Reply: Thank you very much for your suggestion. The written mistake has been revised in the manuscript. Please see page 7, Figure 3C.

(4) On page 4. acetonitrile should be written as CH3CN instead C2H3N illustrating better the real structure of the molecule.

Reply: Thank you very much for your suggestion. The acetonitrile has been expressed by CH3CN in the revised manuscript. Please see pages 4, line 148.

(5) On page 6, line 200, figure should be corrected to Figure.

Reply: Thank you very much for your suggestion. The written mistake has been revised in the manuscript. Please see page 6, line 204.

(6) Sometimes space is added between the value and %, sometimes not. Authors should not write it. For example p. 7. 230, 236 or 237.

Reply: Thank you very much for your suggestion. The written mistake has been revised in the manuscript. Please see pages 7, lines 237, 242-244.

(7) Please improve the quality of EDX image in Figure 3, on the insert showing part of the corresponding bandgap determined by Tauc plot (UV-vis absorption spectrum).

Reply: Thank you very much for your suggestion. The quality of image has been improved in the manuscript. Please see Figure 3.

(8) On page 7, coil should be changed to E. coli (three times). On p. 10, line 306. E. coli should be written with italics.

Reply: Thank you very much for your suggestions. The written mistakes have been revised in the manuscript. Please see pages 7 and 10, lines 235, 249, 251 and 315.

All things considered, this manuscript is a well-written text and showed a valuable preparation method with possible future photocatalytic application in visible-light catalytic sterilization. Therefore, after minor revision, the manuscript can be published in this journal.

Thank you very much for your comments.

Round 2

Reviewer 2 Report

The authors have done the changes I asked for. 

I have detected some more syntax mistakes and I ask the authors to take those into account. Please see the attached file for the corrections. 

Otherwise, the article can be published after these minor corrections.

Author Response

Comments and Suggestions for Authors: The authors have done the changes I asked for. I have detected some more syntax mistakes and I ask the authors to take those into account. Please see the attached file for the corrections. Otherwise, the article can be published after these minor corrections.

Thanks very much for your comments and suggestions. The manuscript has been edited again according to your suggestions. Please see pages 3, 4, 6 8 and 11 of the revised manuscript.
